# Peer review of "The Case for the Development of a Chagas Disease Vaccine: Why? How? When?"

_tropicalmed, 2021, doi:10.3390/tropicalmed6010016_

Round 1

Reviewer 1 Report

This is an interesting essay. However it does not bring relevant knews to the field as vaccination against T. cruzi is still a big challenge, as it is for malaria and leishmaniasis. It was stressed by Authors that the acute phase should be the ideal phase to respond to a vaccine. However this is unpractical to be developed. 

It is interesting to note the evolution of the disease. The cardiac and digestive forms usually only appear many years after the infection, when no blood parasites are detected, but the antibody levels against T. cruzi are present. 

Other point to be observed is that in endemic Chagas disease countries a striking decrease in incidence is noted by combating the transmission vector bugs. Even with the considerations raised by the Authors it is mandatory to compare the disease control already proved effective by the vector control with the still uncertain development of vaccine. 

There is not a vaccine  that has proven to be therapeutically effective in patients with chronic Chagas disease, so far. The potentially preventive vaccine might be important in acute Chagas disease, but the logistic to a widespread vaccination needs to be proposed and evaluated.

Finally, in general, chagasic patients live far bellow the desirable levels of housing and hygiene conditions  in endemic areas. Possibly governmental actions to improve those living conditions should be enough to better protect the citizens against Chagas and other infectious disease as compared to an expensive and innefective vaccine in the light of current knowledge. 

Author Response

This is an interesting essay. However it does not bring relevant knews to the field as vaccination against T. cruzi is still a big challenge, as it is for malaria and leishmaniasis. It was stressed by Authors that the acute phase should be the ideal phase to respond to a vaccine. However this is unpractical to be developed. 

ANSWER: We thank the reviewer for his/her appreciation. We agree that vaccine development against T. cruzi is still a challenge, and in the section entitled “Challenges and the way forward” we review some of the issues faced and propose strategies for future work. We did not propose the acute phase as the ideal phase for vaccination, quite the contrary, we stated as one of the challenges that: “most studies have focused on assessing short term vaccine efficacy (acute phase), which is unlikely to be of relevant clinical use, and it is unclear how these results can translate into long term efficacy”. Thus, we stressed the need for more studies assessing vaccine efficacy in the chronic phase, in agreement with the proposed vaccine product profile.

It is interesting to note the evolution of the disease. The cardiac and digestive forms usually only appear many years after the infection, when no blood parasites are detected, but the antibody levels against T. cruzi are present. 

ANSWER: We agree with the reviewer that this is an important point, and we now stress the clinical manifestations develop many years after infection (Page 2, line 46).

Other point to be observed is that in endemic Chagas disease countries a striking decrease in incidence is noted by combating the transmission vector bugs. Even with the considerations raised by the Authors it is mandatory to compare the disease control already proved effective by the vector control with the still uncertain development of vaccine. 

ANSWER: We certainly agree with the reviewer that vector control has allowed to strongly decrease incidence in most endemic region, but it is also clear that vector control alone is insufficient to eliminate Chagas disease as a public health issue. Our point is that this adds to the rationale for looking for complementary/additional tools for disease control, including vaccines (Page 1, lines 38-42).

There is not a vaccine  that has proven to be therapeutically effective in patients with chronic Chagas disease, so far. The potentially preventive vaccine might be important in acute Chagas disease, but the logistic to a widespread vaccination needs to be proposed and evaluated.

ANSWER: It is correct that vaccines have not yet been found effective in patients so far, because no clinical trials have yet been performed. However we argue that the many preclinical studies showing effectiveness in animal models are encouraging and clinical trials should be considered in the near future to assess safety and efficacy in humans.

Finally, in general, chagasic patients live far bellow the desirable levels of housing and hygiene conditions  in endemic areas. Possibly governmental actions to improve those living conditions should be enough to better protect the citizens against Chagas and other infectious disease as compared to an expensive and innefective vaccine in the light of current knowledge. 

ANSWER: Housing improvement and social programs to reduce poverty are indeed important components to improve health and reduce the incidence of many diseases, but economic studies do show a benefit of having a vaccine against T. cruzi of even moderate efficacy (Page 2, lines 69-72, 74-76, and 94-95). Future clinical studies should help evaluate the level of efficacy that can be reached.

Reviewer 2 Report

The work has scientific merit in reviewing all existing vaccines but it has a serious failure to fail to detail the effectiveness and efficiency of these existing vaccines.

Author Response

Reviewer #2

The work has scientific merit in reviewing all existing vaccines but it has a serious failure to fail to detail the effectiveness and efficiency of these existing vaccines.

ANSWER: We thank the reviewer for his/her appreciation. Comparing the immunogenicity and efficacy of the multiple vaccines tested would indeed be of interest to help select the best candidates for further development. However, such comparisons are not feasible among studies because of the diversity of methods to measure the immune response (and the multiple immune parameters) and the diversity of infection models (mouse and parasite strains, doses of infection, follow-up times, etc...). This is now pointed out (Page 3, lines 113-114).

Reviewer 3 Report

Authors review status and present their opinion regarding Chagas disease vaccine. Overall, it's a well-written review manuscript with clear and sound insight. Here are some comments.

  1. The last paragraph author summary challenges for developing Chagas disease vaccine. It could be better if authors can share some possible experiments or plan to overcome or address those challenges-be more constructive.
  2. The title is not clear. Why? How? When? of doing what? Developing a vaccine or deploying it? Since this is a scientific journal, readers are not just general audience. A more specific title is preferred. 

Author Response

Authors review status and present their opinion regarding Chagas disease vaccine. Overall, it's a well-written review manuscript with clear and sound insight. Here are some comments.

ANSWER: We thank the reviewer for his/her appreciation.

  1. The last paragraph author summary challenges for developing Chagas disease vaccine. It could be better if authors can share some possible experiments or plan to overcome or address those challenges-be more constructive.

ANSWER: We did include some suggestions of future directions for research in the section entitled “Challenges and the way forward” (Page 6), that include the pre-clinical evaluation of long-term vaccine efficacy in the chronic phase, the evaluation of parasite burden in multiple tissues, considerations on parasite diversity and multiple infection, and the need to initiate clinical trials in humans.

  1. The title is not clear. Why? How? When? of doing what? Developing a vaccine or deploying it? Since this is a scientific journal, readers are not just general audience. A more specific title is preferred. 

ANSWER: As suggested, we have revised the tile to be more precise. It now reads: “The case for the development of a Chagas disease vaccine: Why? How? When?”

Round 2

Reviewer 2 Report

The modifications were satisfactory.